# Challenges and Innovations Brought about by the COVID-19 Pandemic Regarding Medical and Pharmacy Education Especially in Africa and Implications for the Future

**DOI:** 10.3390/healthcare9121722

**Published:** 2021-12-13

**Authors:** Ayukafangha Etando, Adefolarin A. Amu, Mainul Haque, Natalie Schellack, Amanj Kurdi, Alian A. Alrasheedy, Angela Timoney, Julius C. Mwita, Godfrey Mutashambara Rwegerera, Okwen Patrick, Loveline Lum Niba, Baffour Boaten Boahen-Boaten, Felicity Besong Tabi, Olufunke Y. Amu, Joseph Acolatse, Robert Incoom, Israel Abebrese Sefah, Anastasia Nkatha Guantai, Sylvia Opanga, Ibrahim Chikowe, Felix Khuluza, Dan Kibuule, Francis Kalemeera, Ester Hango, Jennie Lates, Joseph Fadare, Olayinka O. Ogunleye, Zikria Saleem, Frasia Oosthuizen, Werner Cordier, Moliehi Matlala, Johanna C. Meyer, Gustav Schellack, Amos Massele, Oliver Ombeva Malande, Aubrey Chichonyi Kalungia, James Sichone, Sekelani S. Banda, Trust Zaranyika, Stephen Campbell, Brian Godman

**Affiliations:** 1Department of Medical Laboratory Sciences, Faculty of Health Sciences, Eswatini Medical Christian University, P.O. Box A624, Swazi Plaza, Mbabane H100, Eswatini; etta5013@gmail.com; 2Department of Pharmacy, Faculty of Health Sciences, Eswatini Medical Christian University, P.O. Box A624, Swazi Plaza, Mbabane H100, Eswatini; folarinamu@gmail.com; 3Unit of Pharmacology, Faculty of Medicine and Defence Health, Universiti Pertahanan Nasional Malaysia (National Defence University of Malaysia), Kem Perdana Sungai, Besi, Kuala Lumpur 57000, Malaysia; runurono@gmail.com; 4Department of Pharmacology, Faculty of Health Sciences, Basic Medical Sciences Building, Prinshof Campus, University of Pretoria, Arcadia 0083, South Africa; natalie.schellack@up.ac.za (N.S.); werner.cordier@up.ac.za (W.C.); 5Strathclyde Institute of Pharmacy and Biomedical Sciences, University of Strathclyde, Glasgow G4 0RE, UK; amanj.baker@strath.ac.uk (A.K.); angela.timoney2@nhs.scot (A.T.); 6Division of Public Health Pharmacy and Management, School of Pharmacy, Sefako Makgatho Health Sciences University, Garankuwa, Pretoria 0208, South Africa; moliehi.matlala@smu.ac.za (M.M.); hannelie.meyer@smu.ac.za (J.C.M.); 7Department of Pharmacology, College of Pharmacy, Hawler Medical University, P.O. Box 178, Erbil 44001, Iraq; 8Department of Pharmacy Practice, College of Pharmacy, Qassim University, Buraidah, Qassim 51452, Saudi Arabia; aarshiedy@qu.edu.sa; 9Chair SIGN Council, Healthcare Improvement Scotland, Edinburgh EH12 9EB, UK; 10Department of Internal Medicine, Faculty of Medicine, University of Botswana, Gaborone, Botswana; mwitajc@ub.ac.bw; 11Department of Medicine, Sir Ketumile Masire Teaching Hospital, Gaborone, Botswana; godfreyrwegerera@gmail.com; 12Effective Basic Services (eBASE) Africa, Ndamukong Street, Bamenda, Cameroon; okwen@ebaseafrica.org (O.P.); lumnyanga@gmail.com (L.L.N.); 13Adelaide University, Adelaide 5005, Australia; 14Department of Public Health, University of Bamenda, P.O. Box 39, Bambili, Cameroon; 15Department of Psychology, Faculty of Applied Social Sciences, Eswatini Medical Christian University, P.O. Box A624, Swazi Plaza, Mbabane H100, Eswatini; baffourboahenb@gmail.com; 16Department of Social Work, Faculty of Applied Social Sciences, Eswatini Medical Christian University, P.O. Box A624, Swazi Plaza, Mbabane H100, Eswatini; fcity.tabi@gmail.com (F.B.T.); funkeamu@gmail.com (O.Y.A.); 17Pharmacy Directorate, Cape Coast Teaching Hospital (CCTH), Cape Coast, Ghana; joezy_35@yahoo.com (J.A.); robertincoom30@yahoo.com (R.I.); 18Pharmacy Practice Department of Pharmacy Practice, School of Pharmacy, University of Health and Allied Sciences, Volta Region, Ghana; isefah@uhas.edu.gh; 19Department of Pharmacology & Pharmacognosy, School of Pharmacy, University of Nairobi, P.O. Box 19676-00202 KNH, Nairobi 00200, Kenya; aguantai@yahoo.com; 20Department of Pharmaceutics and Pharmacy Practice, School of Pharmacy, University of Nairobi, P.O. Box 19676-00202 KNH, Nairobi 00200, Kenya; sopanga@uonbi.ac.ke; 21Pharmacy Department, Kamuzu University of Health Sciences (KUHeS), Blantyre, Malawi; ichikowe@medcol.mw (I.C.); fkhuluza@medcol.mw (F.K.); 22Department of Pharmacy Practice and Policy, Faculty of Health Sciences and Veterinary Medicine, University of Namibia, Windhoek 10000, Namibia; dkibuule@unam.na (D.K.); fkalemeera@unam.na (F.K.); ehango@unam.na (E.H.); jlates@unam.na (J.L.); 23Department of Pharmacology and Therapeutics, Ekiti State University, Ado-Ekiti 362001, Nigeria; joseph.fadare@eksu.edu.ng; 24Department of Medicine, Ekiti State University Teaching Hospital, Ado-Ekiti 362001, Nigeria; 25Department of Pharmacology, Therapeutics and Toxicology, College of Medicine, Lagos State University, Ikeja, Lagos 21266, Nigeria; olayinka.ogunleye@lasucom.edu.ng; 26Department of Medicine, Lagos State University Teaching Hospital, Ikeja, Lagos 101233, Nigeria; 27Faculty of Pharmacy, The University of Lahore, Lahore 54000, Pakistan; xikria@gmail.com; 28Discipline of Pharmaceutical Sciences, School of Health Sciences, University of KwaZulu-Natal, Durban 4041, South Africa; OOSTHUIZENF@ukzn.ac.za; 29Pharmaceutical Industry, Pretoria 0002, South Africa; Gustav.Schellack@ppdi.com; 30Hurbert Kairuki Memorial University, 70 Chwaku Road Mikocheni, Dar Es Salaam, Tanzania; amos.massele@hkmu.ac.tz; 31Department of Child Health and Paediatrics, Egerton University, Nakuru, Kenya; ombevaom@gmail.com; 32East Africa Centre for Vaccines and Immunization (ECAVI), Namela House, Kampala, Uganda; 33Department of Pharmacy, School of Health Sciences, University of Zambia, Lusaka, Zambia; chichokalungia@gmail.com; 34Department of Biomedical Sciences, University of Zambia, Lusaka, Zambia; jmaimbos@gmail.com; 35Department of Medical Education Development, University of Zambia, Lusaka, Zambia; ssbanda2007@gmail.com; 36Department of Medicine, University of Zimbabwe College of Health Sciences, Harare, Zimbabwe; zaranyikatrust@gmail.com; 37Centre for Primary Care and Health Services Research, School of Health Sciences, University of Manchester, Manchester M13 9PL, UK; stephen.campbell@manchester.ac.uk; 38NIHR Greater Manchester Patient Safety Translational Research Centre, School of Health Sciences, University of Manchester, Manchester M13 9PL, UK; 39Centre of Medical and Bio-Allied Health Sciences Research, Ajman University, Ajman 20550, United Arab Emirates

**Keywords:** Africa, COVID-19, cross country learning, e-learning, hybrid approaches, Internet platforms, mental health, medical education, pharmacy education

## Abstract

Background: Multiple measures introduced early to restrict COVID-19 have dramatically impacted the teaching of medical and pharmacy students, exacerbated by the lack of infrastructure and experience with e-learning at the start of the pandemic. In addition, the costs and reliability of the Internet across Africa pose challenges alongside undertaking clinical teaching and practical programmes. Consequently, there is a need to understand the many challenges and how these were addressed, given increasingly complex patients, to provide future direction. Method: An exploratory study was conducted among senior-level medical and pharmacy educators across Africa, addressing four key questions, including the challenges resulting from the pandemic and how these were dealt with. Results: Staff and student members faced multiple challenges initially, including adapting to online learning. In addition, concerns with the lack of equipment (especially among disadvantaged students), the costs of Internet bundles, and how to conduct practicals and clinical teaching. Multiple activities were undertaken to address these challenges. These included training sessions, developing innovative approaches to teaching, and seeking ways to reduce Internet costs. Robust approaches to practicals, clinical teaching, and assessments have been developed. Conclusions: Appreciable difficulties to teaching arising from the pandemic are being addressed across Africa. Research is ongoing to improve education and assessments.

## 1. Introduction

Currently, the highest prevalence worldwide for both infectious diseases and non-communicable diseases (NCDs), as well as patients with multiple co-morbidities, including both infectious and non-infectious diseases, is found in Africa [1,2,3,4,5,6]. The prevention and management of these patients, especially those with multiple co-morbidities, will require an improved understanding of the causal determinants of individual and overlapping diseases, including potential genetic factors [7]. Alongside this, there can be co-payment issues in a number of African countries, which can also have catastrophic consequences on patients and their families [8,9,10,11,12]. Consequently, healthcare professionals (HCPs), especially physicians and pharmacists, across Africa need to be aware of potential multiple co-morbidities among their patients and their subsequent impact on patient management, as well as the costs of medicines prescribed, to improve outcomes; however, this is not always the case [13].

There are also growing concerns with high levels of inappropriate prescribing and dispensing of antimicrobials across Africa [14,15,16,17,18,19,20,21,22]. Unless addressed, this will increase antimicrobial resistance (AMR) rates, increasing morbidity, mortality, and costs [17,23,24,25]. However, the availability of trained pharmacists, coupled with the awareness of current regulations, can reduce inappropriate dispensing of antibiotics without a prescription, as seen in Kenya and Namibia [17,26,27,28], similar to other low- and middle-income countries (LMICs) [29]. For this to routinely happen, though, pharmacy education needs to be strengthened across Africa [14,30,31]. This is particularly important at this time since concerns with inappropriate prescribing and dispensing of antimicrobials across Africa have been enhanced by the COVID-19 pandemic [9,32]. Published studies suggest that 70% or more of patients with COVID-19 have received antibiotics even when not clinically indicated [33,34,35,36,37,38]. This is because less than 10% of patients with COVID-19 appear to have either fungal or bacterial co-infections warranting an antimicrobial, driving up AMR rates [39,40,41].

Alongside this, we are aware that the early instigation of preventative measures for COVID-19 across Africa has appreciably reduced routine immunisation among children, driving up future morbidity and mortality [42,43]. Lockdown and other measures, including services paused at clinics, have also increased NCD morbidity and mortality across countries [43,44,45,46]. Community pharmacists can play a key role in addressing these concerns by making vaccination services routinely available in pharmacies, ensuring patients receive their prescribed medicines on time, providing routine health checks, and instigating pertinent interventions to enhance future adherence to prescribed medicines [47,48,49]. This, though, will require appropriate training to fully implement.

Similar to other countries and continents, the pandemic has not only seriously affected people’s lives but also disrupted traditional educational approaches across Africa [43,50]. Among higher education institutions across Africa, there has been a considerable disruption of the academic calendar alongside the temporary closure of institutions. This has resulted in a stressful situation for teaching staff, parents, and students [51], which needs to be addressed going forward.

Adequate training of physicians and pharmacists is essential to deal with increasingly complex cases across Africa given the increasing prevalence of diseases and patients with co-morbidities [7,52,53,54,55]. This is seen as particularly important during the COVID-19 pandemic given the extent of misinformation that exists regarding possible medicines for the prevention and treatment of COVID-19 and their consequences [9,56]. In addition, as mentioned, the extent of inappropriate prescribing and dispensing of antibiotics for viral infections which increases AMR, necessitating pragmatic approaches to address this [17,39].

However, advancing medical and pharmacy education during the pandemic has been a major challenge across countries including African countries [50,57,58]. In the midst of the COVID-19 pandemic, universities across Africa had to respond quickly by offering online classes/courses and practical sessions for healthcare students, and these often had to be offered in smaller groups. The need for practical skills training for physicians and pharmacists is essential to enable them to perform their clinical roles, recognising that delivering both didactic and practice learning remotely is challenging. This meant that both faculty members and students had to firstly become familiar with new technologies and platforms if not already familiar with them, and faculty members needed to deliver lectures and practicals safely whilst preserving the quality and consistency of educational processes as much as possible. Students also needed to cope with issues of Internet connectivity and access to computers to fully participate in e-learning approaches [59,60,61,62,63,64,65,66,67]. The financial challenges that this situation imposed among both staff members and students across Africa were quite considerable. This was in addition to the lack of suitable equipment and facilities for online learning among many African students, especially disadvantaged students, certainly initially [68] (Table 1).

Agreed standards are also increasingly needed to enhance online learning experiences given the current challenges, including successfully transferring courses and assessments online [61,69,70,71]. Alongside this, as mentioned, there must be good availability and access to the Internet and appropriate software packages, as well as availability of computers among students, especially those from disadvantaged backgrounds, to facilitate e-learning [50,61,62,71,72,73]. However, this was not always the case (Table 1, Appendix A with Appendix A containing more details for each location). Key challenges across Africa at the start of the pandemic included the need to rapidly improve staff skills to transfer courses online, the lack of suitable technology equipment among students, and the costs and availability of the Internet, as well as how to conduct practicals within COVID-19 restrictions (Table 1).

We have seen similar responses in high-income countries, including Saudi Arabia in the neighbouring region to Africa, in terms of closing medical and pharmacy schools during the early months of the pandemic and suspending all face-to-face classes. However, medical and pharmacy education in Saudi Arabia has been facilitated by significant investment in e-learning management systems and technological innovations over the years, providing direction to others [74]. Box 1 contains further details of ongoing activities across Saudi Arabia to enhance physician and pharmacy education during the COVID-19 pandemic. We are also seeing a rise in the use of technologies such as telemedicine or remote education platforms to address skill concerns among healthcare students across countries, and this will grow [75].
Box 1Activities across Saudi Arabia to enhance e-learning and address the challenges posed by the COVID-19 pandemic.There has been considerable investment in recent years in the e-learning infrastructure among Saudi Universities. This includes the establishment of Deanships/Centres of e-learning and distance education, enhancing a smooth transition during the pandemic [74,76].Alongside this, e-learning management systems, online platforms, and other technological infrastructures, e.g., BlackBoard™, online apps, and platforms for collaboration and video calls, as well as other technological infrastructures put in place, including Zoom^®^, Microsoft Teams, and Webex^®^ [74,76,77,78].In 2020–2021, the Saudi Ministry of Education instructed universities to adapt a blended learning model in which all theoretical components of the curriculum should be delivered online and practical component/skills delivered on campus whenever applicable, taking all the precautionary and preventive measures [79]. This resulted in the King Saudi University (KSU) College of Medicine delivering the theoretical parts of courses online using BlackBoard™ and Zoom^®^ platforms, with problem-based learning (PBL) shifted online using small groups [77].Practical sessions have been classified into three categories depending on the most suitable mode of delivery—these include on-campus sessions in the labs and virtual sessions via online platforms or video demonstrations uploaded in the e-learning systems. For clinical teaching, more priority has been given to final-year medical students to be trained at the KSU medical city, while for other years, training and education has been conducted mainly in the Clinical Skills and Simulation Centre (CSSC) at the college with provisions to provide adequate orientation to the faculty members and students regarding educational and training activities to ensure smooth running of the programme [77].In the case of pharmacy students, the use of videos has been explored to provide demonstrations of experiments, especially during the closure of colleges in the initial months [71]. This was subsequently re-visited after restrictions were eased, with the adoption of a blended learning model similar to the model for medical education.The safe return to university campus education was implemented under the theme “Cautiously We Return” with a key aim to implement the practical and clinical components of the curriculum as well as examinations on campus—facilitated by adopting digital health technologies and applications introduced in the early phases of the pandemic to mitigate risks, e.g., all members (faculty, students and other employees) in all Saudi universities had to have an activated Tawakkalna app on their smartphones to enable entry to the universities. The Tawakkalna app is an official app and includes many advanced digital services with a high level of security and privacy features, with the app showing the current health status of the users (i.e., infected, not infected), any recent contact with an infected individual, and their COVID-19 vaccination status (i.e., fully vaccinated, vaccinated with the first dose, not vaccinated) based on Ministry of Health data through encrypted personal data [80].Several studies have now taken place exploring the experiences of university students. They have generally reported positive experiences with e-learning and distance education during the COVID-19 pandemic in view of the investments made [71,78,79,81,82], with continued ongoing research in this area. However, this is not universal, with concerns with technical support and access to the Internet found among some medical students in Saudi Arabia [83].

We have also seen advances in e-learning approaches among a number of countries, including higher-income countries, to devise different teaching approaches. These include mobile applications for practical sessions, addressing concerns with e-exams, and universities combining to offer virtual microscopy and Zoom videoconferencing to teach pathology [84,85,86,87]. The situation in these countries, as well as Saudi Arabia (Box 1), can be different to experiences in Jordan and other LMICs with appreciably fewer resources to instigate e-learning and other approaches [50,88,89,90], as well as across Africa (Table 1).

Consequently, given the many challenges facing medical and pharmacy training in Africa during the pandemic, including pre-clinical activities [91] (Table 1), coupled with the urgency of the situation given the increasing complexity of patients across Africa, we believed it was necessary to document ongoing activities among African countries to improve medical and pharmacy education during the pandemic and the lessons learnt.

We were aware of a number of innovations across Africa during the early stages of the COVID-19 pandemic to improve patient care. These included developing finger-prick tests to rapidly detect antibodies; developing mobile apps for contact tracing, triaging, and case management; early sequencing of the genomes of SARS-CoV-2; and the development of non-invasive respirators given initial global shortages [43,92]. We now wanted to document initiatives regarding medical and pharmacy education during the pandemic across Africa, given that e-learning and other digital technologies are here to stay [93,94,95]. We believe this is the first time that such activities have been reviewed across a continent rather than individual countries, which should add strength to any future recommendations. The objective being to facilitate countries, especially African countries, to learn from each other [91,96].

We also recognise the need for students and faculty members to continually re-evaluate the changes in their educational approaches and assessment processes as the pandemic continues in order to effectively address continued challenges [97]. Alongside this, there is a need to adapt approaches to address identified gaps including improving self-regulated behaviour, with students looking out for each other [98,99,100,101,102,103]. We further recognise that preceptors should be open-minded and consider how students can participate in service provision, such as clinical pharmacy services, during pandemics [104]. This includes a greater use of virtual communication, including potentially with patients, access to electronic health records remotely where these exist to enhance analysis and treatment skills, and greater flexibility in teaching methods [104,105].

As a result, we believed it was important that there was increased understanding of the “new normal” in moving forward as a continent educating future physicians and pharmacists. This is seen as essential not only in the current pandemic but also considering the possibilities for future pandemics. Consequently, this paper sets out to understand the many challenges faced by senior-level personnel from different universities across Africa, particularly in medical and pharmacy education, as a result of the COVID-19 pandemic, and how these have been dealt with, in order to share lessons learnt and the implications for the future to improve student education. This also includes suggested research activities for the future.

## 2. Materials and Methods

This was principally an exploratory study among senior-level medical and pharmacy educators across Africa during the current COVID-19 pandemic, using an analytical framework with a pragmatic paradigm approach to provide future direction [106,107]. The African countries chosen provided a range of geographies, economic status (Gross Domestic Product—GDP/capita [108]), and population size [109] (Table 2).

Similar approaches have previously been adopted when documenting activities across Africa to combat both infectious and non-infectious diseases [4,9,14,43,110,111,112].

The questionnaire was developed following an analysis of the literature regarding key points, concerns, and activities with respect to changes in medical and pharmacy education resulting from lockdown and other measures to combat the COVID-19 pandemic (Table 1, Appendix A). Four key questions were contained in the questionnaire to address the aims and objectives of the study. These were:What challenges has COVID-19 presented to health sciences education?How did health sciences institutions respond immediately to the challenges presented by the COVID-19 pandemic in your country (principally medical and pharmacy education)?What support was harnessed to help mitigate against the challenges faced by higher learning institutions?What are the lessons that can be learnt to prepare higher learning institutions in the education of particularly physicians and pharmacists for future pandemics?

The questionnaire was subsequently distributed to senior-level co-authors in each participating country using a purposeful sampling approach [113]. The co-authors subsequently collated the replies based on their experiences, local reports, and internal institutional consultations in their own countries, most of which they were heavily involved with. In some countries, including Ghana, Nigeria, South Africa, and Zimbabwe, multiple institutions were approached with the findings initially collated and reviewed by the principal author (B.G.) with the help of the co-authors from that country. The replies to the questions were discussed among the multiple co-authors and the principal co-author until a consensus was reached. In countries with only one principal university teaching medicine and pharmacy, only one university was approached, e.g., Botswana, Eswatini, and Namibia. The lead author in that country typically collated the comments from the multiple co-authors; alternatively, the co-authors built and refined their comments, building on others in that country. As a result, a consensus was reached among all participating countries. This reflects the exploratory nature of this study coupled with its aims and objectives.

The responses were analysed using thematic analysis techniques [114,115]. Common themes across Africa in response to the four key questions were identified and subsequently collated by the co-authors, combined with other colleagues in some cases, to provide future direction (Figure 1). These built on the comprehensive answers to the four questions by the senior-level co-authors among the various African countries.

## 3. Results

The COVID-19 pandemic has resulted in considerable challenges, barriers, and shifts in the way that medical and pharmacy education are now practised among universities across Africa (Figure 2). We will split the findings down into the four questions before consolidating the findings to provide future guidance.

### 3.1. What Challenges Has COVID-19 Presented to Health Sciences Education?

Table 3 provides details from respondents about the range of challenges they faced regarding the education of medical and pharmacy students. Challenges across Africa included the need for both staff and students to quickly adapt to online learning, concerns with the lack of available equipment, and the costs of Internet bundles. Alongside this, there were concerns with how to undertake practical sessions, clinical teaching, and examinations during lockdown restrictions alongside instigating problem-based learning approaches. These built off challenges discussed in Table 1. Appendix A provides greater details for each African country.

### 3.2. How Did Health Sciences Institutions Respond Immediately to the Challenges Presented by the COVID-19 in Your Country (Principally Medical and Pharmacy Education)?

Table 4 summarises the various ways in which the universities across Africa immediately responded to the challenges in medical and pharmacy education as a result of lockdown and other measures, with greater details in Appendix A.

This typically included extensive use of remote teaching involving the education of both staff and students. In addition, adjustments were made to the teaching schedules to take into account the new environment, as well as adjustments to practical and teaching programmes to take into account social distancing and other regulations.

### 3.3. What Support Was Harnessed to Help Mitigate against the Challenges Faced by Higher Learning Institutions?

Table 5 discusses the support that was harnessed by the various institutions across Africa in response to the pandemic. This included entering into agreements with Internet providers to help with the costs of e-learning approaches for medical and pharmacy students, as well as improved sanitation and hand-washing facilities. Some universities also provided psychological and moral support to staff and students to help address ongoing challenges.

### 3.4. What Are the Lessons That Can Be Learnt to Prepare Higher Learning Institutions in the Education of Physicians and Pharmacists for Future Pandemics?

Table 6 discusses the lessons that have been learnt regarding the future education of medical and pharmacy students during a pandemic and beyond. This includes the fact that theoretical teaching can be smoothly taught virtually, with blended learning here to stay. However, this needs investment in skills and equipment, as well as increasing the Internet reliability and coverage and reductions in the costs of Internet bundles. Institutions also need to be made safe for future pandemics, and there needs to be greater psychological and other support for both students and staff going forward. Medical and pharmacy institutions also need to continually develop and adapt pertinent tools for practicals, clinical teaching, and examinations to address current challenges.

### 3.5. Consolidated Findings and Their Implications

A number of common themes were identified regarding the impact of the COVID-19 pandemic on medical and pharmacy education across Africa as well as potential ways forward. These build on Table 3, Table 4, Table 5 and Table 6 and Box 1. They are consolidated in Table 7 to provide guidance to key stakeholders in Africa and the wider world experiencing similar problems with physician and pharmacy education.

## 4. Discussion

There is no doubt that the COVID-19 pandemic, along with measures to slow its spread across Africa—incorporating early lockdown alongside other measures, including closure of borders and higher education institutions [43,92]—has caused considerable challenges for higher education across Africa (Table 1). This is similar to many other countries [50,85,86,87,98,118].

We believe this is the first time that the implications of these multiple measures on higher institution learning, especially for medical and pharmacy students, and ways forward have been assessed across a single continent, Africa, with its considerable challenges before the start of the COVID-19 pandemic. We believe this was important given the many public health and other challenges across Africa, which include the increasing complexity of patients exacerbated by the growing number of patients with multiple co-morbidities and the challenges this brings, such as the need to consider multiple guidelines simultaneously to appropriately manage these complex cases [55,119,120,121,122]. Alongside this, concerns with the lack of ICT infrastructures and e-learning facilities among many universities in African countries before the start of the pandemic (Table 1).

Concerns with the education of physicians and pharmacists across Africa arising from the pandemic were exacerbated by the initial challenges of translating lectures to online, the lack of familiarity with online e-learning platforms among many faculty members and students, and, as mentioned, poor ICT infrastructures among many African countries (Table 1 and Table 3). This is similar to other LMICs, including Jordan [50,88,123]. There was also a considerable unplanned financial burden among both universities and students at the start of the pandemic with the need to purchase computers, tablets, and other ICT support systems necessary for e-learning, as well as the cost of Internet bundles and user fees among staff and students. These factors affected all key stakeholders involved in HCP education across Africa at the start of the pandemic. This included governments, non-governmental organisations, and the private sector, as well as the institutions, staff, and students (Table 1 and Table 3). The situation regarding medical and pharmacy education at the start of the pandemic was different, though, in Saudi Arabia, with greater infrastructure initially with its greater resources (Box 1). However, despite this, there were still a number of issues that needed addressing in Saudi Arabia at the start of the pandemic, including the optimal blending of online and face-to-face teaching, with the findings from recent studies likely to influence future teaching approaches (Box 1).

Table 3 and Table 7 document the many barriers and challenges faced by universities and higher education institutions across Africa at the start of the pandemic, especially for medical and pharmacy students. These include addressing concerns with the lack of infrastructure and competency in e-learning approaches. In addition, there were concerns about shortages of ICT equipment, especially among disadvantaged students, as well as the cost of Internet bundles. Alongside this, coping with learning at home given competing demands and the availability of quiet places to teach and learn were significant concerns. There were also concerns with conducting practicals and clinical teaching sessions with social distancing and other measures in place, including WASH, which necessitated breaking students down into smaller groups [77]. In addition, there were concerns with the need for greater investment in simulation-based learning (SBL) although countries are learning from each other to address this.

Table 4, Table 5 and Table 7 discuss the different responses among the various institutions across Africa to address these multiple challenges. These included instigating training seminars, webinars, and other types of development platforms for both staff and students, along with developing innovative approaches to e-learning, including the use of videos, turning existing university electronic media into learning management systems, and exploring the potential for hybrid teaching sessions. Whilst this may have incurred initial logistical challenges within universities, faculty development programmes have allowed for more certainty in how to approach new modalities, stimulated the development of future practice approaches, and provided opportunities for professional development [124,125,126]. Alongside this, we are aware of innovative approaches to the training of surgeons in addition to those documented (Table 4, Table 5, Table 6 and Table 7). These include enhancing video platforms and online teaching of instrument identification, suturing, and knot tying [127,128,129].

By allowing for the continuous development of academics’ educational toolboxes, opportunities arise not only for engaging with new fields of research but also for enhancing preparedness for future disruptions, including future pandemics. Capacity building was also important among countries given concerns with the lack of investment over a number of years among some of the African universities at the start of the pandemic, which are now being addressed. Flexibility was also seen as key, including calendar adjustments, given the different circumstances among the students, with some students struggling to adjust to online learning. Governments and universities also need to work more closely in the future with Internet providers to address concerns with the cost of Internet bundles.

Encouragingly, developments in e-learning and other approaches are taking place across Africa to address the many highlighted challenges (Table 4, Table 5 and Table 7). New ways to assess students have also been developed whilst seeking ways to maintain the robustness and integrity of the assessment systems. This is because security and rigour of assessments were noted as a general concern among academics in Africa early on in the pandemic, mirroring other countries [86]. These concerns are largely due to the necessary transition to online platforms, which cannot always be invigilated, and the requirement that physical assessments must be conducted outside of the authentic environment during any pandemic, e.g., objective-structured clinical assessments. Although institutions have done much to implement a variety of quality control processes, mistrust in the process is still evident in a number of African countries, and future research is needed to assess the effectiveness of different strategies to address remaining fears and provide future guidance. We will continue to monitor this. Other potential research activities emanating from African university activities during the pandemic are discussed in Table 8.

Overall, different approaches have been instigated to ensure medical and pharmacy students are equipped as much as possible with the necessary skills at graduation (Table 5 and Table 7). This includes a greater need for evidence-based treatment approaches given the discourse surrounding a number of different treatments for patients with COVID-19, including hydroxychloroquine and remdesivir, that failed to improve patient care and, in some cases, actually increased morbidity, mortality, and costs [9,32,56,130,131,132,133,134]. In addition, there is a greater need for effective ways to reduce unnecessary prescribing and dispensing of antibiotics, including in patients with COVID-19, as well as to improve the management of patients with NCDs given concerns with the unintended consequences of the pandemic [17,27,43,44,135].

**Table 8 healthcare-09-01722-t008:** Potential research activities to improve the education of medical and pharmacy students during the current and future pandemics.

Potential Research Projects
Determine the costs associated with capacity building to improve online teaching, including greater support for staff and students. Assess the costs and effectiveness of instigating WASH (water, sanitation, and hygiene) facilities throughout universities to enhance safety for both students and staff. This could include infrastructure costs where there are concerns with social distancing and other hygiene measures. Continually assess the costs and effectiveness of different hybrid models for teaching medical and pharmacy students to encompass agreed designated core competencies. This includes different approaches to practicals and clinical sessions—including the potential for small groups and monitoring any adjustments to subsequent teaching/practical approaches, where pertinent, as well as their subsequent effectiveness to provide future guidance. This to take account of a greater cognisance of the potential difficulties and challenges within home environments for both staff and students alongside the challenges with ensuring appropriate skills at graduation, including, for instance, surgical skills, as well as the potential for increased flexibility in learning approaches. Explore the potential for changing primary research modules into secondary research modules including systematic reviews, and whether such approaches can achieve agreed educational objectives. Develop tools that help monitor the quality of teaching with hybrid models as well as assess the quality of online assessments, given enhanced potential for plagiarism, and implement them. Continue to evaluate different approaches that enhance the robustness of online assessments, including examinations, that necessarily replace formal examinations during current and future pandemics, and seek to implement the most appropriate approaches. Continue to evaluate the costs and effectiveness of different psychological, moral, and counselling support for both staff and students during a pandemic, especially if this interferes with learning, and implement appropriate recommendations. Research strategies that help mitigate against future pandemics through analysing current case histories across Africa and seek to help implement the findings. We have seen African countries learn from each other, and this is likely to continue [43]. Explore the potential for joint pan-African collaboration and research into key research activities surrounding the education of medical and pharmacy students during pandemics given the complexities of managing patients in Africa, especially those with joint infectious and non-infectious diseases, building on previous joint research projects [4,111,112,136,137].

Table 6 and Table 7 summarise key learnings arising from the challenges posed by the pandemic and how these can be addressed going forward. This includes the fact that theoretical learning can in fact be taught virtually, with this blend of e-learning and face-to-face teaching for medical and pharmacy students here to stay. This pandemic and its consequences have also provided universities across Africa with the potential to re-think their syllabi to include hybrid learning opportunities. However, this will require the necessary infrastructure, equipment, and personnel to effectively take this forward. Activities potentially include establishing e-learning centres within universities where these currently do not exist, developing quality standards for e-learning, providing adequate technological resources, i.e., online platforms, and ensuring the necessary training for faculty members and students. This will require greater collaboration between technology firms within Africa, governments, and the universities, with greater corporate responsibility among companies [73,138]. Internet bundles also need to be affordable to all for hybrid approaches to work in the future. We will also likely see greater collaboration among universities in the future alongside continual research into appropriate teaching methods for this and future pandemics (Table 8). We will also be monitoring this.

The pandemic, including the development of innovative management and treatment approaches [43] coupled with innovative approaches to teaching, has also provided an opportunity for African universities to showcase their skills to attract future investment. We have already seen apps being developed in South Africa to rapidly assess the use of antimicrobials among patients in hospitals given current concerns [17,110,139,140], with such approaches likely to grow, building on the greater use of technologies to manage patient care across Africa and the wider world [43,75,141,142,143].

Innovative ways could also be introduced, including digital solutions, to provide increasing psychological and social support and guidance for staff and students during the pandemic given current concerns [51,65,66,144,145]. The objective being to reduce anxiety, depression, and other mental health issues associated with the pandemic. This is because challenges regarding the mental health and well-being of staff and students was a common observation across Africa (Table 3). Mental health issues among staff and students were generally linked to concerns including uncertainty with the pandemic in terms of altered working conditions. Alongside this, difficulties with maintaining a work–life balance; fear of the implications of the transition to e-learning on teaching, learning, and assessments; and the loss of face-to-face support networks, with burnout of students and lecturers an ongoing concern [146,147,148]. However, we have seen the growth of virtual support networks across universities as well as increased compassion towards one another to help address such issues. This is likely to continue.

We are aware of a number of limitations with this study. These include the fact that we only approached a limited number of universities and other personnel involved in HCP education in each country and no students. In addition, we did not cover all African countries. We also did not undertake a thorough thematic analysis of the answers to the four questions for the reasons stated. However, in view of the seniority of the co-authors and the insights they provided, we believe our findings are robust, providing direction for the future.

## 5. Conclusions

In conclusion, the pandemic has required higher education institutions across Africa to dramatically change the way in which they approach the training of their HCP students. This includes appreciably increasing hybrid approaches incorporating e-learning coupled with face-to-face teaching where permissible. Alongside this, instigating innovations to help with practicals and clinical teaching, as well as with assessments to ensure their quality and robustness. We have also seen key stakeholder groups come together to address concerns with ICT support for students, especially disadvantaged students, as well as the costs of Internet bundles, with such collaborative approaches likely to grow. Addressing mental health concerns among both students and staff has become increasingly essential during any pandemic, and given the experiences seen during the current pandemic, this will continue.

As the world moves further into the Fourth Industrial Revolution, paradigm shifts will continue to occur. As COVID-19 has shed light on the weaknesses and strengths of educational institutions across Africa and in many cases amplified them, the unique challenges each institution has faced with training their medical and pharmacy students to become fit-for-purpose graduates have become increasingly clear. We have seen different institutions successfully adopt a variety of strategies to address these challenges, with hybrid learning here to stay. Consequently, it is unlikely that institutions across Africa will revert to the same educational strategies that were in place prior to the COVID-19 pandemic, especially with continued investment and research into e-learning opportunities and systems. We will continue to monitor the situation to provide future guidance.

## Figures and Tables

**Figure 1 healthcare-09-01722-f001:**
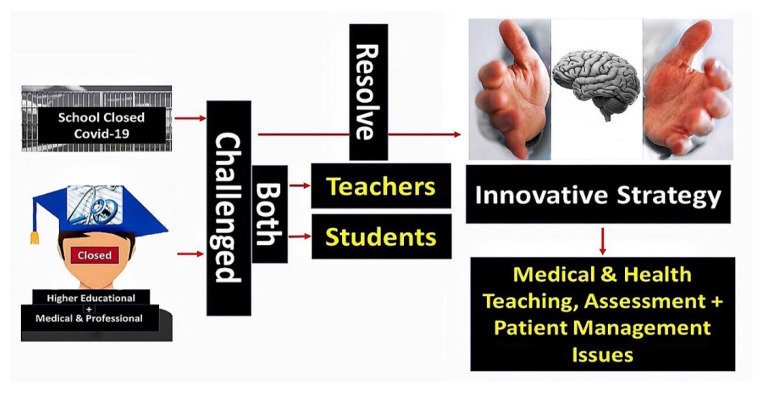
Key challenges and future opportunities arising from the COVID-19 pandemic.

**Figure 2 healthcare-09-01722-f002:**
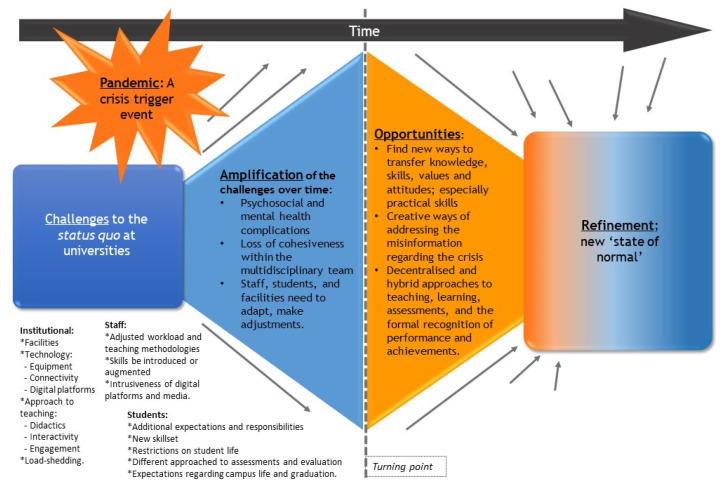
COVID-19 and the impact on medical and pharmacy education across Africa.

**Table 1 healthcare-09-01722-t001:** Summary of challenges and activities among faculty and students across Africa in response to the COVID-19 pandemic.

Location	Challenges				Activities		
	Transferring Courses Online	Staff Skills at the Outset of the Pandemic	Reliability/ Cost of Internet	Concerns with Practical Skills Learning/Good Communication between Students and Staff	Rapid Adaptation to e-Learning, Including Virtual Platforms/ Sourcing ICT Devices	Providing Positive Learning Experiences/Good Communication with Students	Adapting Frameworks/ Additional Security Measures for Assessing Students
North Africa	√		√		√	√	√
East Africa			√	√		√	
Central Africa	√			√	√		
Southern Africa	√	√	√	√	√	√	√
Western Africa	√	√	√	√	√		

NB: ICT = Information and communications technology.

**Table 2 healthcare-09-01722-t002:** Current GDP/capita and population size among participating African countries.

Country	GDP/Capita (USD)	Population Size
Malawi	625.3	19,129,952
Uganda	817.0	45,741,007
Zambia	1050.9	18,383,955
Tanzania	1076.5	59,734,218
Zimbabwe	1128.2	14,862,924
Cameroon	1499.4	26,545,863
Kenya	1838.2	53,771,296
Nigeria	2097.1	206,139,589
Ghana	2328.5	31,072,940
Eswatini	3415.5	1,160,164
Namibia	4211.1	2,540,905
South Africa	5090.7	59,308,690
Botswana	6711.0	2,351,627

**Table 3 healthcare-09-01722-t003:** Summary of key challenges faced by universities and students across Africa as a result of the pandemic.

Country	Teaching and Learning Challenges						Resource Issues
	Disruption of the Academic Calendar/ Limited Experiences with e-Learning, Shortages of Staff	Lack of Suitable ICT Equipment, Literacy, and the Need for Hybrid Learning	Lack of Reliable Internet Facilities/ Challenges with Home Learning	Concerns with Practical Skills Learning as Well as Face-to-Face Learning	Instigation of Problem-Based Learning to Address Lack of Face-to-Face Learning	Quality Assessment/ Robustness of Examinations	Cost of Internet/ Facilities
Botswana	√	√	√		√		√
Cameroon	√		√	√			√
Eswatini	√	√	√	√		√	√
Ghana	√		√	√		√	√
Kenya	√	√	√	√		√	√
Malawi	√	√		√			√
Namibia	√	√	√	√	√	√	√
Nigeria	√	√	√	√	√		√
South Africa	√		√	√		√	√
Uganda	√	√	√	√		√	√
Zambia	√	√	√	√		√	√
Zimbabwe	√	√	√	√			√

**Table 4 healthcare-09-01722-t004:** Summary of immediate responses among institutions across Africa to the pandemic and its impact on educational approaches for physicians and pharmacists.

Country	Responses to Teaching			Responses to Practicals and Clinical Teaching	
	Extensive Use of e-Learning Approaches, Including Remote Teaching and Blending/ Training of Staff	Regular Adjustments of Academic Calendars/ Teaching Schedules	Instigation of Social Distancing/ Sanitation Measures for Lectures	Modification of Practicals/Contact between Students and Patients	Adaptive Approaches, Including Systematic Reviews in Small Groups/Interactive Videos/Use of Workbooks
Botswana	√	√		√	√
Cameroon	√		√		
Eswatini	√		√	√	√
Ghana	√		√	√	√
Kenya	√	√	√	√	√
Malawi	√		√	√	
Namibia	√	√	√	√	√
Nigeria	√	√	√	√	√
South Africa	√	√	√	√	√
Zambia	√	√		√	√
Zimbabwe	√	√	√	√	

**Table 5 healthcare-09-01722-t005:** Examples of support and other measures harnessed by health science institutions across Africa in response to the pandemic.

Country	Support
Botswana	The University of Botswana entered an agreement with one network provider (Botswana Telecommunications Corporation) to provide 1 GB every day to students. The Ministry of Education provided PPE for the students.
Cameroon	Provision of water points with soap for hand washing among institutions. Display of COVID-19 sensitization materials around medical and pharmacy school campuses. Provision of hand sanitisers and face masks for attendance in lectures.
Eswatini	Capacitation of teaching staff was provided by the university through training on the use of selected online teaching and learning platforms. Training on preventive measures against COVID-19 was provided to safeguard staff and students during blended learning sessions once lockdown measures were eased. Staff were provided with face masks and face shields to enhance their face-to-face teaching. The university made arrangements to provide psychosocial, moral, and spiritual support to students and staff affected by the pandemic. Telecommunication companies including MTN Eswatini, Eswatini Mobile, and Eswatini Post and Telecommunication (EPTC) provided subsidies on data bundle packages to assist students and staff with Internet costs.
Ghana	Enhancement of existing university electronic media into a learning management system that enhanced lecturer and student interaction. Introduction of teaching and student online assessments and re-assessments. Students who performed poorly during their assessments were given a remedial assignment to make up for the negative effect of switching to virtual learning.
Kenya	Provision of Internet bundles to students and staff, as well as subscription by the University of Nairobi to online teaching platforms. Formation of COVID-19 committees that were tasked with carrying out research to inform practice as well as education of staff and students about the pandemic. Capacity building of the university ICT department to provide adequate support services and training for online teaching during the pandemic. Conducting online graduation ceremonies for students who completed their studies. Counselling services provided to students and staff to reduce COVID-19 related anxiety. Partnerships with corporate organisations for raising funds to provide PPE for students and staff, as well as donations of PPE to staff and students. Building more hand-washing stations in the universities as well as provision of clean water and soap to all staff and students to reduce the spread of the virus.
Malawi	Out of the many strategies, the blended learning strategy has been widely used, as it allows those that are in quarantine to be able to teach and learn. Blending of the use of the COMPASS platform with other platforms such as Zoom^®^, Google Classroom, and WhatsApp^®^. Blended learning helps with issues of crowding in classes, as all big classes are being taught using the system, as well as helping those students that failed to attend online learning sessions to cover missed sections and improve their understanding of the various concepts. Conducting blended graduation and thesis defence with online and face-to-face meetings. Only students with first class/distinction went for physical graduation, while the rest graduated online. Introduction of hand sanitiser production facility in the Pharmacy department with the aim of providing quality hand sanitisers to the staff and students of the College and University. Provision of ICT equipment (computers and tablet) to students to help them access online platforms. Initiatives with Internet service providers that changed the providers from purely a private entity status to a consortium managed by institutions of higher learning in Malawi. This resulted in doubling of Internet bandwidth to improve connectivity to both the staff and students.
Namibia	The University of Namibia, through its Centre for Online, Distance and e-Learning (CODeL), capacitated academics and students to engage in online learning, teaching, and assessments. The university provided various remedial opportunities for teaching and assessments for students whose learning was adversely affected by the COVID-19 pandemic. The Ministry of Higher Education established a fund to enhance research on COVID-19 as well as purchase ICT devices for underprivileged students. A flexible working schedule was permitted by the university, including the use of various teaching methods and online platforms. Together with other stakeholders, the University of Namibia also provided free Internet dongles to students and heavily subsidised Internet dongles among staff members to address perceived challenges. Some electronic books were made available free of charge to students by the university library.
Nigeria	Listening/viewing hubs were created in some institutions to enable students who could not afford Internet subscriptions to gather for lectures within the campuses. Some sub-national governments were able to provide additional financial resources and infrastructure to aid online learning among tertiary institutions. Sharing of ideas and innovative approaches to learning in the pandemic era and working towards bridging the learning inequality gaps through professional bodies and societies. Some institutions supported their academic staff financially to sustain online teaching, with some universities also providing free Wi-Fi for lecturers’ online academic activities.
South Africa	Concerns with the readiness of staff and students to use online teaching platforms was addressed through several workshops for both students (as end-users) and teaching staff for implementation of alternative pedagogies. It was recognised that the development of online material needed to consider the various learning styles of the different students, including differences that may be present among ethnically diverse student bodies. Continuous webinar distribution of lectures alongside educational units helping to offer expedited training—however, this did create a sense of overload/not doing enough/failing to attend everything among some students. Videoconferencing software offered the ability to divide students among different “breakout rooms” for team-based learning. Online discussion boards stimulated discourse on individual topics but were less well-suited for complex topics such as patient case discussions. Establishment of counselling services or reinforcing current services to support and strengthen the mental health and well-being of students and staff. Several agreements between universities, the government, and Internet providers for negotiating zero-rated access to specific educational and information websites and data bundles for staff and students to facilitate Internet access. Some universities also obtained a greater GSuite licence to help accommodate a greater integration between Gmail and other GSuite platforms (which was eventually downgraded due to cost).
Tanzania	Online communication is now encouraged in public and private universities whenever possible. Increasing support to help address financial constraints and Internet connectivity among staff and students.
Zambia	COVID-19 prevention policies and statutory provisions were enacted to help reduce the spread of the virus. The universities endeavoured to orient their faculty to re-acquaint them to teaching using online platforms. Negotiations took place with Internet service providers to address concerns. The result was the zero-rated deal with MTN Zambia, a mobile service provider. In this regard, MTN Zambia collaborated with the University of Zambia and Copperbelt and Mulungushi Universities (who are the main public universities offering medical programmes in Zambia) to provide subsidized access to Internet in order to enable students to study from home during the COVID-19 pandemic [116]. However, issues of universal access to Internet services remain relatively inequitable and characterised by low uptake, which needs to be addressed going forward.
Zimbabwe	Establishing free Internet connectivity at the University of Zimbabwe College of Health Sciences for students as well as some teaching hospitals. Continued face-to-face teaching for clinical rotations, mindful of the virus and the implications Rotation of didactic lessons and clinical sessions to enhance learning during the pandemic.

NB: ICT = Information and communications technology; PPE = Personal protective equipment.

**Table 6 healthcare-09-01722-t006:** Lessons learnt from the pandemic among institutions providing medical and pharmacy education across Africa.

Country	Lessons Learnt
Botswana	Theoretical teaching can smoothly be taught virtually, provided that students and staff have access to the Internet. However, strong Internet and electronic platforms must be established. For medical training, the hands-on experience needs to be considered and limits for any mandatory contact time experience need to be established. Welfare services for staff and students need to be established to address issues of anxiety and depression brought on by the pandemic and its consequences.
Cameroon	For online learning or distance learning using digital platforms to be effective, Internet connectivity needs to improve in bandwidth and speed at an affordable cost. It is possible that 5G technology may be needed for Cameroon to progress in this direction. Water, sanitation, and hygiene (WASH) issues pose a risk to educational attainment at all levels of learning in Cameroon, not only in the face of the COVID pandemic but also in the face of other epidemics and outbreaks, including cholera and Ebola. Consequently, educational institutions must invest in WASH for improved learning and teaching outcomes in the future.
Eswatini	The COVID-19 pandemic put a considerable strain on the majority of the higher learning institutions, including health science institutions, who had depended on conventional face-to-face teaching and learning approaches, and compromised the mandate of tertiary institutions. The challenges encountered and lessons learnt in preparation for future pandemic include: ○ Capacity building for health science institutions in the use of online learning platforms, as well as integration of virtual learning environments into institutional websites, in order to normalise remote learning; ○ There is need for health science institutions to strengthen collaboration with major stakeholders, including the Eswatini Higher Education Council as well as telecommunication- and technology-based companies to reinforce a stronger infrastructure to support remote learning, thereby enhancing quality education during future pandemics; ○ The psychological and emotional effect of the COVID-19 pandemic suffered by both staff and students in health science institutions suggests the need to develop a formal psychosocial support system in every training institution to cater for students and staff whose academic responsibilities may be compromised as a result of such a pandemic.
Ghana	The need to develop remote and distance teaching and learning materials have provided the opportunity to rethink the curricula, as well as teaching and learning processes and the restructuring of health science students’ competencies assessments for future pandemics. The need for partnerships to develop or enhance the virtual capacity of health science education in the area of skill-based learning and assessment. The need to develop targeted support for students and lecturers to address the barriers and challenges in providing quality education during an outbreak. Health science universities must also collaborate to develop policies on virtual teaching and learning during an outbreak. They should also focus on scaling up or developing the relevant capacity to utilise resources available to them when developing or producing mitigatory measures in future pandemics.
Kenya	Going forward, universities must invest in proper infrastructures that can enhance social distancing and hygiene measures during pandemics. Universities also need to invest heavily in ICT to support online teaching, research, and other related activities. There is a need for recognising the promotion of blended learning (face-to-face and online) and less emphasis on physical learning. Alongside this, a need to invest in alternative methods of clinical teaching, such as the use of videos and demonstrations. Using COVID-19 as a case history to conduct research and develop strategies for mitigating future pandemics.
Malawi	Need to develop full-time online lessons whereby students can choose either online or face-to-face learning. This requires institutions to have fully developed and up-to-date online learning systems and infrastructures. Institutions need to fully develop online resources and regularly update them accordingly for use by current students so that they can be fully developed and up-to-date and more inclusive in advance of any future pandemic. Medical and pharmacy institutions also need to develop pertinent tools for virtual experimentation of practicals to address current challenges.
Namibia	There is need for health science institutions to migrate to education appropriate for the 4th Industrial Revolution (Education 4.0/5.0), which is learner-centred, work-integrated, and competence-based and embraces digital education through a range of technologies [117]. Directly transplanting materials used for face-to-face teaching to online platforms does not achieve the same results; students’ comprehension of topics after online lectures (whether recorded or synchronous) was not as high as after the same lectures were given face-to-face. Consequently, additional effort needs to be made to ensure that: Students attend synchronous teaching sessions with lecturers; Lecturers actively engage as much of the class as possible during these sessions to keep students focussed and address any misunderstandings. Hands-on practicals and clinical training of health science students should never be substituted with 100% online learning and assessments, although there can be challenges during any pandemic. There is need to assure the continued quality of online assessments as they are prone to plagiarism. Health science institutions in resource-limited settings need to continuously innovate and improve teaching, learning, and assessments in their local context to enhance appropriate training of students. Developing learning materials (such as question banks, case studies, and demonstrations) in collaboration with other health science institutions across Africa will strengthen the quality of future training while reducing duplication of effort.
Nigeria	Establishment of functional hybrid learning platforms (in-person and online) in educational institutions. Development of reliable virtual online assessment methods for students. Adequately equipped skills acquisition laboratories among the universities, especially public universities, to bridge the gaps created by the non-feasibility of physical learning in future pandemics. Educational curricular reviews to integrate emerging and re-emerging diseases into the medical and pharmacy educational programmes. The need for health sciences regulatory bodies to ensure acceptable standards are achieved to guarantee the quality and competence of HCPs who graduate during pandemics.
South Africa	There have been ample learning opportunities for all involved as well as a greater appreciation for the need for institutions to exercise compassion, flexibility, and, where possible, a sense of stability for students. Everyone, including staff and students, was forced to be innovative, think more creatively, and come up with appropriate solutions for alternative ways and approaches to ensure teaching and learning takes place according to pre-set criteria. As a result, better preparation for future pandemics. Considering the advantages of hybrid or blended learning as an educational approach for future medical and pharmacy education, building on existing institutional strengths. Online teaching and learning opened up endless possibilities of collaboration between academic institutions locally and abroad to share expertise and include teaching by international experts as part of the curricula, which was previously not possible. The importance of clear communication and expectations. Many students were not aware of what the environment would be like or the need to continually adapt assessments to accommodate online learning. Greater need for a scholarship of teaching and learning in academic training to ensure alignment throughout, which also provided ample opportunity to engage in HCP education research.
Tanzania	Both public and private universities should be strongly encouraged to invest in appropriate infrastructures that can enhance social distancing and hygiene during pandemics. Universities also need to invest heavily in ICT to support online teaching, research, and other related activities. The future for teaching and learning in Africa will be based on IT to deliver lectures, assignments, and examinations, the latter being a considerable challenge that needs urgent addressing.
Uganda	Uganda needs to invest in the infrastructure of universities to offer more hybrid learning experiences going forward. Faculty members need to be more flexible with greater e-learning in the home environment and the challenges this produces. Universities need to work with the government and Internet providers to reduce the financial burden of Internet access and platforms for working at home.
Zambia	Universities must invest in robust e-learning educational strategies and accessible educational media technologies for health science education in the future. E-learning interventions for medical and pharmacy education could be of great benefit for targeting quality education, with over-reliance on face-to-face, contact-based teaching and learning approaches revisited going forward. The effectiveness of e-learning, including blended learning approaches, needs to be evaluated in the context of core competence development and fitness for subsequent practice for the graduates produced using such methods, which is ongoing in Zambia. In this digital era, increasing uptake of e-learning and making quality Internet accessible at subsidised cost (inexpensive) for student populations must be a mainstream strategy for universities to enhance access to learning platforms.
Zimbabwe	There is a need to establish reliable, affordable digital platforms for e-learning among both university staff and students. There is also a need for universities and colleges to collaborate with tech companies for potential solutions to enhance e-learning opportunities. Build student capacity to become citizen programmers so that they become tech-savvy as part of the curriculum and guide others. Universities need to pay more attention to psychosocial support for students in future pandemics.

NB: ICT = Information and communications technology; HCP = Healthcare professionals.

**Table 7 healthcare-09-01722-t007:** Common themes regarding the impact of challenges of the COVID-19 pandemic on teaching practices and the lessons learnt.

Question	Common Themes
Challenges faced by universities during the pandemic	Rapid assimilation of new teaching methods and platforms by both staff and students to facilitate e-learning and transition away from traditional face-to-face approaches. Concerns with issues of digital literacy with the different technologies and platforms among both staff and students, as well as infrastructure concerns within universities to deal with online/virtual learning. In addition, shortages of the basic essential resources to deal with online/ virtual learning both at the institutional and individual (lecture and student) level, especially for disadvantaged students. The rapid transformation of face-to-face teaching to online which needed to take into account key issues, including diversity in terms of race, cultural identities, language, and socioeconomic backgrounds, as well as challenges with conducting problem-based learning online. Concerns with access to reliable Internet facilities (coverage, cost and reliability) for both staff and students, as well as concerns with the effectiveness of mobile technologies with regard to issues of Internet access and costs. Challenges with conducting practicals and electives for medical and pharmacy students and sourcing materials. This included providing the necessary PPE and ensuring social distancing for any sessions conducted in universities after the initial lockdown. Educational challenges due to a shortening of the academic year following the initial wave of the pandemic. In addition, challenges faced by staff and students in their own home environments coping with competing demands. Difficulty with data collection for any primary research as part of assessments for graduation Barriers and challenges with developing new methods to maintain the quality and integrity of student assessments through virtual platforms. Coping with the disruption of staff/student exchange programmes as well as visiting lecturers. Mental health and performance concerns of students and staff as a result of the pandemic and its uncertainties, including extended teaching hours of staff members.
Responses by health science institutions to the pandemic	Webinars and training sessions have been conducted to enhance the skills of both staff and students towards e-learning/online learning platforms as part of the transition from face-to-face learning to virtual online approaches. This included the provision of “Internet buddies” in some universities. Researching the potential for hybrid teaching sessions as lockdown measures eased and some students wanted to return to campus. Continually re-assessing the quality of teaching to address misconceptions/concerns with current approaches. This included the potential for increased flexibility with regards to particular modules to allow for different home circumstances. Introduction of formative assessments online undertaken through appropriate platforms to ensure robustness/integrity. Developing innovative approaches, including videos, dividing students into small groups, demonstrating practicals online (via YouTube and other approaches), and using problem-based learning approaches. Videoconferencing software offered the possibility of dividing students into different “breakout” sessions. As lockdown measures eased, practical sessions and clinical students in wards were staggered with fewer students at a time and with students provided with full protective equipment. Changing of primary research modules as part of courses into secondary research including systematic reviews. Working with Internet access providers to address concerns with the purchasing of data bundles by staff and students, as well as seeking innovative approaches to make laptops and other equipment readily available to students, especially disadvantaged students. Providing PPE to students and staff when on campus and encouraging regular hand washing/hand sanitisation.
How support was harnessed to help mitigate against the pandemic	Turning existing university electronic media into learning management systems, enhancing lecturer and student interaction, alongside transforming lectures online. Forming COVID-19 groups within universities tasked with introducing new learning approaches as well as assessing the appropriateness of current approaches/refinements where necessary. This also included suggestions for capacity building of IT infrastructure within universities/additional support for staff and students to address barriers to the use of new platforms/approaches, as well as the instigation of Internet buddies in some universities. Providing flexibility to students who were struggling, especially those struggling from any negative effects arising from the switch to online learning in the home environment. Ensuring as far as possible that online assessments were robust to maintain the quality of degree graduates and conducting graduation ceremonies online. Harnessing support from governments and Internet providers to assist with the cost of access to the Internet. Similarly, seeking ways to provide ICT devices to students, including computers, where finances were an issue, e.g., disadvantaged students. Giving assistance to staff and students to support online learning as well as generally to address mental health and other issues arising from the pandemic.
Lessons learnt and ways forward	Theoretical teaching can be taught virtually, provided that staff and students have access to suitable platforms, devices and reliable Internet facilities. However, this requires appropriate inputs initially to build up skilled-based learning approaches in new electronic media, including capacity building in a number of institutions. The move to remote learning/blended learning provided the opportunity for universities to re-think the syllabi and future approaches away from principally traditional face-to-face teaching approaches. This though, may require universities to invest in appropriate infrastructures/media technologies and re-think how to appropriately conduct practicals and clinical teaching sessions in the future, as well as ensure clear communication and expectations of all involved. Online teaching and learning opened up possibilities for greater collaboration between academic institutions locally and abroad, as well as sharing expertise and involving international experts in teaching programmes, which was not always possible before. The pandemic and its implications also highlighted the need for university staff to continually re-evaluate their teaching approaches as well as examination approaches to ensure expectations are met and graduates are up to agreed standards, especially given the increasing complexity of patients in Africa. Alongside this, it is important to re-think the organisation of clinical rotations and placements, to reduce the number of students in one clinical setting at a time, while still maintaining quality of learning. Universities need to ensure students and staff have access to appropriate ICT devices and Internet bundles—especially disadvantaged students, as this was a problem initially among many medical and pharmacy students across Africa. This could include encouraging more corporate responsibility among companies and individuals as well as working with governments and other support systems. Universities also need to ensure the well-being of staff and students going forward with changes in workloads and locations, e.g., home environments. Overall, the re-thinking of teaching approaches, coupled with other innovations seen across Africa as a result of the pandemic [43], provides considerable opportunities for African universities to showcase their skills to attract future investments; in other words, COVID-19 could be the catalyst for the next revolution with its implications for the future of higher education across Africa and the wider world.

## Data Availability

All available data are included in the paper.

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
