# Peer review of "Challenges and Innovations Brought about by the COVID-19 Pandemic Regarding Medical and Pharmacy Education Especially in Africa and Implications for the Future"

_healthcare, 2021, doi:10.3390/healthcare9121722_

Round 1
Reviewer 1 Report
Manuscript ID: healthcare-1432219
Title:
Challenges and innovations brought about by the COVID-19 pandemic regarding Medical and Pharmacy education especially in Africa and implications for the future
The present manuscript was nicely described. It provides clear and detailed information on the struggle faced by Medical and Pharmacy educators in Africa during the present pandemic. Most importantly, the study provides insights on how to dealt with future ‘the new normal’ in education.
I enjoy reading the introduction as it comprehensively describes presents both past and current situation of Medical and Pharmacy education in Africa. It gives a clear and sufficient background to understand the problem under investigation. The last paragraph of introduction section highlights the importance of the study, and it is in line with the other parts of the manuscript.
The Materials and Methods section is straightforward with clear explanation of collecting data process. The tables give detailed information that helps the reader understand the situation in each country.
The discussion section sums up the study well. It gives insight on how we face ‘the new normal’, especially in an educational setting. It also appropriately describes the importance of other aspects in education beside syllabus and curriculum, such as equipment, internet access, and the price of internet bundles. The discussion delivers the study result in a comprehensive, yet insightful way.
Overall, I am very glad to read this nicely written and needed study.
My feedback to improve the manuscript:
Please provide a concise information in Table 1, 3 and 4. The tables are too long to read and it drifts the reader’s attention away. Moreover, some of the information overlapped between countries.
My suggestion would be to explain the similar situation in paragraphs and the difference in tables.
Another suggestion is to summarize the result in a checkbox table.
Author Response
Manuscript ID: healthcare-1432219
Title: Challenges and innovations brought about by the COVID-19 pandemic regarding Medical and Pharmacy education especially in Africa and implications for the future
A) The present manuscript was nicely described. It provides clear and detailed information on the struggle faced by Medical and Pharmacy educators in Africa during the present pandemic. Most importantly, the study provides insights on how to dealt with future ‘the new normal’ in education.
I enjoy reading the introduction as it comprehensively describes presents both past and current situation of Medical and Pharmacy education in Africa. It gives a clear and sufficient background to understand the problem under investigation. The last paragraph of introduction section highlights the importance of the study, and it is in line with the other parts of the manuscript.
The Materials and Methods section is straightforward with clear explanation of collecting data process. The tables give detailed information that helps the reader understand the situation in each country.
The discussion section sums up the study well. It gives insight on how we face ‘the new normal’, especially in an educational setting. It also appropriately describes the importance of other aspects in education beside syllabus and curriculum, such as equipment, internet access, and the price of internet bundles. The discussion delivers the study result in a comprehensive, yet insightful way.
Overall, I am very glad to read this nicely written and needed study.
Author comments: Thank you for your kind words – very much appreciated
B) My feedback to improve the manuscript:
i) Please provide a concise information in Table 1, 3 and 4. The tables are too long to read and it drifts the reader’s attention away. Moreover, some of the information overlapped between countries.
My suggestion would be to explain the similar situation in paragraphs and the difference in tables.
Another suggestion is to summarize the result in a checkbox table.
Author comments: Thank you for this suggestion. As seen, we have now Tabulated these 3 Tables into checkbox tables to provide core components (along with a summary) with the completed picture now available as Supplementary material for those wanting more information on each country. We have not undertaken this for the remaining Tables as this is not suitable. We trust this is now acceptable.
b) We have also gone through the manuscript and made minor changes to help improve the English and flow. We hope this is now acceptable
Reviewer 2 Report
The authors should add discussions of the following sections:
(1) Theoretical and/or analytical framework/s to be used in the recontextualization of the findings/results within relevant literature;
(2) Research paradigm (it looks like the study used Pragmatic paradigm);
(3) Issues of validity;
(4) Sampling (recruitment strategy);
(5) Educational implications.
Author Response
The authors should add discussions of the following sections:
(1) Theoretical and/or analytical framework/s to be used in the recontextualization of the findings/results within relevant literature
Author comments: Thank you – now added into the methodology with references
(2) Research paradigm (it looks like the study used Pragmatic paradigm);
Author comments: Thank you – now added into the methodology with references
(3) Issues of validity;
Author comments: Thank you – as stated in the Methodology the key questions were based on an analysis of the literature across Africa (Table 1 and Supplementary Table 1) and agreed with the senior level co-authors from across Africa and wider. As a result, providing robustness and validity to this approach. We hope this is now acceptable
(4) Sampling (recruitment strategy);
Author comments: Thank you – this was purposeful with a reference now added in
(5) Educational implications
Thank you – We believed we have summarised the educational implications in terms of responses in Table 4 and Supplementary Table 3, the lessons learnt in Table 6, the impact of the pandemic on teaching practices/ lessons learnt in Table 7 and the resultant implications for future research in Table 8. We hope this is acceptable.
We have also been through the manuscript and made minor adjustments to improve the flow and English. We hope this is now acceptable
Reviewer 3 Report
Thank you very much for sending me this manuscript. This is a very well-written article with timely and important implications. I have several comments.
- I appreciate that the authors discuss the limitations that the COVID-19 has brought in terms of clinical experiences. What do studies say about fields like surgery? How can e-learning integrate this aspect of experiences?
- The authors also ‘health science’ in the manuscript. I strongly recommend that the authors stick to medical/pharmacy education.
- For the future research, what kind of data should we collect to further investigate these issues? Please discuss.
Author Response
Thank you very much for sending me this manuscript. This is a very well-written article with timely and important implications. I have several comments.
Author comments: Thank you for your kind words – very much appreciated!
1. I appreciate that the authors discuss the limitations that the COVID-19 has brought in terms of clinical experiences. What do studies say about fields like surgery? How can e-learning integrate this aspect of experiences?
Author comments: Thank you for this. We have included some comments in the Tables 5,6,7 – especially possible future ways forward. We have also made a specific comment on the implications for training surgeons with additional references. We hope this is now acceptable
2. The authors also ‘health science’ in the manuscript. I strongly recommend that the authors stick to medical/pharmacy education.
Author comments. Thank you – now changed. We hope this is now OK
3. For the future research, what kind of data should we collect to further investigate these issues? Please discuss.
Author comments: Thank you for this – we have now consolidated our thoughts into a specific Table 8 and hope this is now acceptable.